# Negative spin Hall magnetoresistance of normal metal/ferromagnet bilayers

Min-Gu Kang[1,5], Gyungchoon Go[2,5], Kyoung-Whan Kim [3], Jong-Guk Choi[1], Byong-Guk Park [1✉] & Kyung-Jin Lee [2,4✉]

Interconversion between charge and spin through spin-orbit coupling lies at the heart of condensed-matter physics. In normal metal/ferromagnet bilayers, a concerted action of the interconversions, the spin Hall effect and its inverse effect of normal metals, results in spin Hall magnetoresistance, whose sign is always positive regardless of the sign of spin Hall conductivity of normal metals. Here we report that the spin Hall magnetoresistance of Ta/NiFe bilayers is negative, necessitating an additional interconversion process. Our theory shows that the interconversion owing to interfacial spin-orbit coupling at normal metal/ferromagnet interfaces can give rise to negative spin Hall magnetoresistance. Given that recent studies found the conversion from charge currents to spin currents at normal metal/ferromagnet interfaces, our work provides a missing proof of its reciprocal spin-current-to-charge-current conversion at same interface. Our result suggests that interfacial spin-orbit coupling effect can dominate over bulk effects, thereby demanding interface engineering for advanced spintronics devices.

[1] Department of Materials Science and Engineering, KAIST, Daejeon 34141, Korea. [2] Department of Materials Science and Engineering, Korea University, Seoul 02841, Korea. [3] Center for Spintronics, Korea Institute of Science and Technology, Seoul 02792, Korea. [4] KU-KIST Graduate School of Converging Science and Technology, Korea University, Seoul 02841, Korea. [5]These authors contributed equally: Min-Gu Kang, Gyungchoon Go. ✉email: bgpark@kaist.ac.kr; kj_lee@korea.ac.kr

A normal metal (NM)/ferromagnet (FM) bilayer is a system of extensive research nowadays as it offers a framework to investigate various spin–orbit coupling (SOC) physics, in particular, the interconversion between charge and spin currents. From the physics point of view, identifying its dominant mechanism is of crucial importance for fundamental understanding of spin and charge transport coupled through spin–orbit coupling[1]. From application point of view, moreover, understanding the interconversion mechanism enables to efficiently manipulate magnetization, which finds use in next-generation magnetic memories[2].

A representative example of the charge-to-spin conversion is the spin Hall effect (SHE) of NM[3]. Owing to the SHE, in-plane charge currents passing through NM are converted into perpendicular spin currents, which exert spin–orbit torques on FM and switch magnetization[4,5]. The Onsager reciprocity states that the inverse SHE converts perpendicular spin currents into in-plane charge currents[6,7], facilitating electrical detection of spin current.

A combined action of the SHE and inverse SHE in NM/FM bilayers causes spin Hall magnetoresistance (SMR); the resistivity changes with the $y$ component of magnetization, where the $y$-axis is perpendicular to both directions of charge-current flow ($x$) and thickness ($z$)[8,9]. In metallic NM/FM bilayers[10,11], the resistivity as a function of magnetization is described as

$$\rho = \rho_0 + \Delta\rho_1 m_x^2 - \Delta\rho_2 m_y^2, \quad (1)$$

where $\rho_0$ is the magnetization-independent resistivity, $\Delta\rho_1$ and $\Delta\rho_2$ are ones for the anisotropic magnetoresistance and SMR, respectively, and $m_x$ ($m_y$) is the normalized magnetization along the $x$ ($y$) direction.

The bulk SHE of NM has been considered as a dominant mechanism for SMR[8–12]. It generates spin current with transverse spin polarization (**y**). This spin current is reflected at the NM/FM interface with an amount depending on the magnetization: more reflected for **m** = **y** than for **m** = **z**. As the reflected spin current reduces net spin current in NM, the charge backflow induced by the inverse SHE of NM is smaller for **m** = **y** than for **m** = **z**, resulting in a positive SMR (i.e., $\Delta\rho_2 > 0$). As the SMR for this mechanism is proportional to $\theta_{SH}^2$, where $\theta_{SH}$ is the spin Hall angle of NM, it is always positive regardless of the sign of $\theta_{SH}$. We note that NM/anti-FM bilayers can exhibit a negative SMR owing to the spin-flop transition[13], which is however absent in NM/FM bilayers.

Another mechanism for the charge-to-spin conversion is the interfacial Rashba SOC that also generates SMR-like magnetoresistance, Rashba magnetoresistance, through current-induced spin density from the Rashba–Edelstein effect[14,15] or anomalous velocity from the band structure modified by Rashba SOC[16–18]. The models for the Rashba magnetoresistance are based on a two-dimensional system without interfacial spin–orbit scattering of the spin current traveling along the thickness direction and predict a positive $\Delta\rho_2$. These models are however incomplete, especially for metallic NM/FM bilayers because direct interfacial generation of spin current flowing in the thickness direction is hindered. It has shown that the spin accumulation from the Rashba–Edelstein effect always generates a positive SMR[14].

Recent first-principles[19,20] and Boltzmann transport[21,22] calculations considering three-dimensional transport found that in-plane charge current is efficiently converted into perpendicular spin current at the NM/FM interface. This interface-generated spin current has been experimentally confirmed[23]. We note that the interfacial spin-current generation is different from the interfacial spin-density generation (i.e., Rashba–Edelstein effect) because the latter is a spin accumulation that generates a spin

current through the spin-diffusion process, whereas the former is a spin current originating from different scattering amplitudes depending on the relative orientation between conduction electron spin and interfacial spin–orbit field. The Onsager reciprocity guarantees that there must be the reciprocal effect, the interfacial conversion from perpendicular spin current to in-plane charge current. A first-principle theory[19] addressed this reciprocal effect through a substantial increase of inverse SHE at the interface. In experiment, however, it is uneasy to disentangle the interface effect from the bulk effect because both have the same symmetry.

In this work, we report negative SMR of Ta/NiFe bilayers, which is opposite to that originating from the existing charge-to-spin conversion mechanisms of the spin Hall (or Rashba–Edelstein) effect. Theoretical analysis shows that the negative SMR is a combined action of the interfacial spin–orbit coupling-induced spin current generation and its reciprocal effect. Our results demonstrate that the interconversion between charge and spin in NM/FM bilayers can be dominated by the interfacial spin–orbit coupling effect, thereby requiring interface engineering for the development of high-efficiency spintronic devices.

## Results

**Angular dependence of the magnetoresistance.** Here we report SMR of Pt(3)/NiFe($t_F$) and Ta(3)/NiFe($t_F$) samples with varying NiFe thickness $t_F$ (in nanometers, see "Methods"). We measure the longitudinal magnetoresistance (MR) $R_{xx}$ with rotating an external magnetic field of 9 T in the $yz$ plane and applying a direct current of 100 μA in the $x$ direction of a Hall-bar structure (Fig. 1a, b). In Fig. 1c–e, we show angle-dependent MRs for Ta/NiFe, Pt/NiFe, and NiFe samples as a function of $t_F$. Similar to the conventional SMR, the MR follows $\Delta\rho_2 \cos^2\beta$, where $\Delta\rho_2 = \rho(\mathbf{m} = \mathbf{z}) - \rho(\mathbf{m} = \mathbf{y})$ and $\beta$ is the polar angle of **m**. In Fig. 1f–h, we plot the SMR ratio $\Delta r_{MR} = \frac{R_{xx}(\mathbf{m}=\mathbf{z}) - R_{xx}(\mathbf{m}=\mathbf{y})}{R_{xx}(\mathbf{m}=\mathbf{z})}$ as a function of $t_F$. The SMR ratio ($\Delta r_{MR}$) of single-layer NiFe samples is negative for all tested $t_F$ ranges (Fig. 1h). The sign of $\Delta r_{MR}$ for Ta/NiFe (Fig. 1f) sample changes at $t_F = 2$ nm, whereas the sign of $\Delta r_{MR}$ for Pt/NiFe (Fig. 1g) sample changes at $t_F = 10$ nm. On the other hand, the angle-dependent MR in other rotating planes ($xy$ and $xz$ planes, see Supplementary Note 1) retain the same sign, irrespective of $t_F$. Therefore, the $t_F$-dependent sign change of NM/FM bilayers is special for the angle-dependent MR in the $yz$ plane, suggesting that $\Delta r_{MR}$ is determined by at least two competing processes with different signs. We note that there is a negligible effect of a MgO(1.6 nm)/Ta(2 nm) capping layer on negative $\Delta r_{MR}$, which is confirmed by examining two control samples: a NiFe(5 nm)/Ta(3 nm)/MgO(1.6 nm)/Ta structure where the stacking order of Ta and NiFe is reversed and a Ta(3 nm)/NiFe(5 nm)/MgO(10 nm)/Ta structure where the MgO layer (10 nm) is thick enough to prevent spin current transmission from the top MgO/Ta interface (Supplementary Note 2).

Known processes that contribute to $\Delta r_{MR}$ are the geometrical size effect (GSE) of FM layer[24,25], the spin Hall (or Rashba-induced) effect in NM (NM/FM interface)[8–18], and anomalous Hall effect (AHE) in FM layer[25]. Among these, the only GSE can give rise to a negative $\Delta r_{MR}$[24,25], whereas others always give a positive $\Delta r_{MR}$[8–18,26]. Our result for the single-layer NiFe samples indeed shows the negative $\Delta r_{MR}$, in agreement with that originating from the GSE. Therefore, we need to check if the GSE of NiFe single layer is sufficient to explain the negative SMR of bilayers, or an additional yet-unknown process is required.

We exclude the GSE as the origin of the negative SMR in Ta/NiFe bilayers for the following reasons. The maximum magnitude of the negative $\Delta r_{MR}$ in Ta/NiFe samples is about twice larger

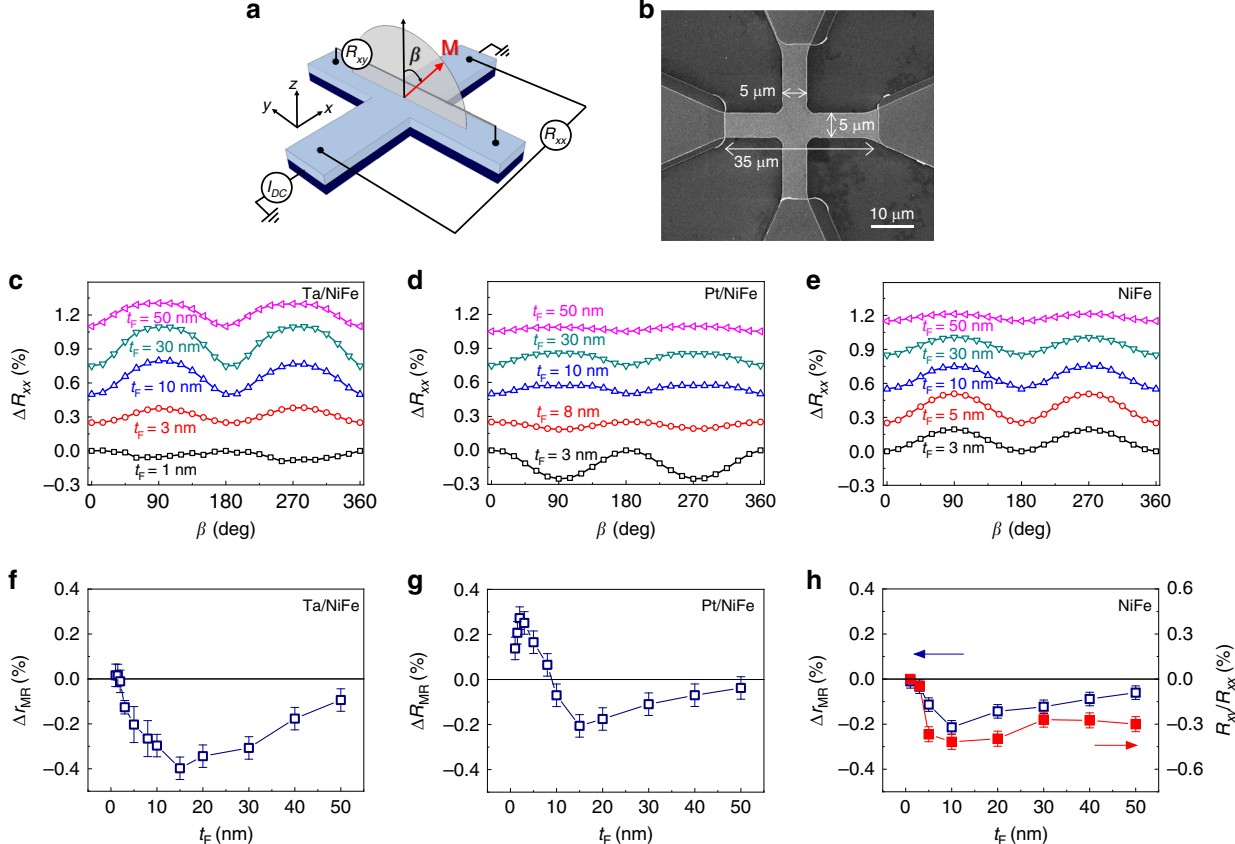

**Fig. 1 Magnetoresistance measurement. a** Schematic illustration of the Hall-bar device for magnetoresistance (MR) measurement and definition of the magnetic field angle $\beta$ in the $yz$ plane. The longitudinal and transverse resistances ($R_{xx}$, $R_{xy}$) are measured using a DC current ($I_{DC}$) of 100 µA. **b** SEM image of the Hall bar. **c–e** $\beta$-angle dependence of the MR [$\Delta R_{xx} = (R_{xx}(\beta) - R_{xx}(\beta = 0))/R_{xx}(\beta = 0)$] in Ta/NiFe sample (**c**), Pt/NiFe sample (**d**), and NiFe single-layer sample (**e**). Results in **c–e** are intentionally offset for clarity. **f–h** Ferromagnet-thickness ($t_F$) dependence of MR ratio ($\Delta r_{MR}$) of Ta/NiFe sample (**f**), Pt/NiFe sample (**g**), and NiFe single-layer sample (**h**). Red squares in **h** show the normalized anomalous Hall resistance (=$R_{xy}/R_{xx}$) of NiFe single-layer sample. The error bars in **f–h** are from the statistical variation of the SMR.

than that of single-layer NiFe samples, i.e., $\left| \Delta r_{MR}^{Ta/NiFe} \right| > \left| \Delta r_{MR}^{NiFe} \right|$. Considering only bulk effects (i.e., the SHE of bulk NM and the GSE of bulk FM) but neglecting the interfacial spin–orbit coupling (ISOC) effect, the parallel circuit model shows that the SMR ratio in NM/FM bilayer is written as $\Delta r_{MR}^{N/F} = \frac{R_0}{R_F} \Delta r_{MR}^F + \frac{R_0}{R_N} \Delta r_{MR}^N$, where $R_F$, $R_N$, and $R_0$ are the magnetization-independent resistance of FM, NM, and NM/FM bilayer, respectively (Supplementary Note 3), and $\Delta r_{MR}^F$ ($\Delta r_{MR}^N$) is the resistance change (in ratio) of FM (NM) layer (see Supplementary Note 4 for details). Because $\Delta r_{MR}$ is positive due to bulk SHE, whereas $\Delta r_{MR}^{N/F}$ and $\Delta r_{MR}^F$ are both negative in the experiment (Fig. 1f, h), inequality $\left| \Delta r_{MR}^{N/F} \right| < \left| \frac{R_0}{R_F} \Delta r_{MR}^F \right|$ must be satisfied. Given that $R_0/R_F < 1$ at a given thickness $t_F$, one finds that $\left| \Delta r_{MR}^{N/F} \right| < \left| \Delta r_{MR}^F \right|$, which contradicts the experimental observation in Fig. 1f, h, i.e., $\left| \Delta r_{MR}^{Ta/NiFe} \right| > \left| \Delta r_{MR}^{NiFe} \right|$. This contradiction shows that there is another origin of the negative $\Delta r_{MR}$, which is not the GSE. Moreover, we investigate the NM thickness dependence of SMR in Ta($t_N$)/NiFe(5 nm) bilayers with Ta thickness $t_N$ ranging from 1 to 12 nm. As seen in Fig. 2, the SMR is negative for all $t_N$'s, and its magnitude becomes maximum when $t_N = 1$–2 nm. The GSE contribution from the NiFe (5 nm) layer (estimated from the parallel circuit model, $\Delta r_{MR}^{N/F} = \frac{R_0}{R_F} \Delta r_{MR}^F + \frac{R_0}{R_N} \Delta r_{MR}^N$ with fixed $\Delta r_{MR}^F$ and $\Delta r_{MR}^N = 0$) is shown by a red line. We find that the

GSE alone cannot explain the negative SMR in Ta/NiFe bilayer, necessitating an additional source of the negative SMR in this sample. We attribute the negative SMR of the Ta/NiFe bilayer to the spin-charge conversion via ISOC.

Assuming that the ISOC is the origin of the negative SMR, a meaningful experimental test is to measure SMR and spin–orbit torque in Cu/NiFe bilayers, where the bulk SHE is negligible. We find that the Cu/NiFe samples exhibit a negative SMR that cannot be explained by the GSE alone (Supplementary Note 5), and also a sizable amount of the damping-like spin–orbit torque (Supplementary Note 6). This result further suggests that the ISOC effect contributes to the negative SMR. A theoretical support for the ISOC as the origin of negative SMR is given in the next section.

**Various charge-to-spin conversion processes**. The negative SMR ($\Delta r_{MR}$) of Ta/NiFe samples cannot be explained by the previously proposed mechanisms [i.e., bulk SHE of NM, interfacial Rashba–Edelstein effect, and AHE], which always give a positive $\Delta r_{MR}$. To figure out a possible origin of the negative SMR, we below list various charge-to-spin conversion processes in NM/FM bilayers. In NM, there is the bulk SHE. In FM, the AHE, planar Hall effect (PHE), and bulk SHE of FM are present[26–28]. Finally, right at the interface[19–23,29], ISOC converts in-plane charge currents into perpendicular spin currents. We here ignore the MR from the current-induced spin density from the Rashba–Edelstein effect or anomalous velocity from the band structure modified by

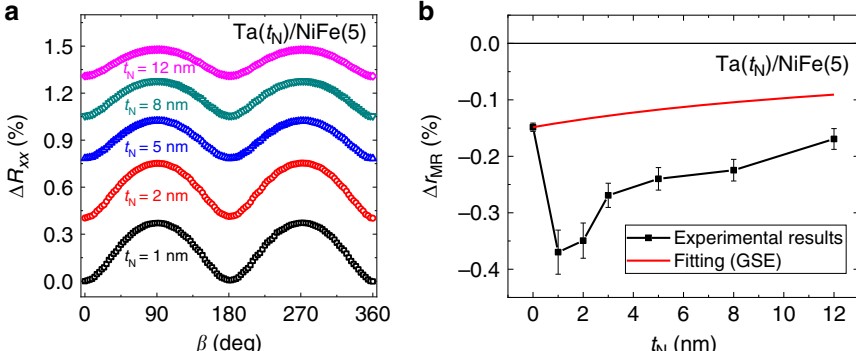

**Fig. 2 Ta-thickness dependence of SMR. a**, $\beta$-angle dependence of the MR [$\Delta R_{xx} = (R_{xx}(\beta) - R_{xx}(\beta = 0))/R_{xx}(\beta = 0)$] in Ta/NiFe samples with various Ta thickness $t_N$. **b**, Ta-thickness ($t_N$) dependence of MR ratio ($\Delta r_{MR}$) of the Ta/NiFe sample. Red line in **b** shows the GSE contribution of the NiFe (5 nm) layer, estimated from the parallel circuit model. The error bar in **b** is from the statistical variation of the SMR.

| Table 1 The sign of MR ratio ($\Delta r_{MR}$) caused by various spin-to-charge conversion processes. | | | |
| --- | --- | --- | --- |
| Charge-to-spin Spin-to-charge | SHE | AHE | ISOC |
| **SHE** | (1) Always positive | (3) Positive for NiFe/Ta bilayers | (5) Canceled by reciprocal effect of (6) |
| **AHE** | (4) Positive for NiFe/Ta bilayers | (2) Always positive | (7) Canceled by reciprocal effect of (8) |
| **ISOC** | (6) Canceled by reciprocal effect of (5) | (8) Canceled by reciprocal effect of (7) | (9) Negative for NiFe/Ta bilayers |

*SHE* bulk spin Hall effect in NM, *AHE* anomalous Hall effect in FM, *ISOC* interfacial spin–orbit coupling effect at NM/FM interface. The numbers in parentheses correspond to those explained in the main text.

Rashba SOC, which does not alter our main conclusion because they only generate positive SMR[14,15].

Among these five processes, the PHE is irrelevant to the $\Delta r_{MR}$ because the PHE is absent for $\mathbf{m} = \mathbf{z}$ and $\mathbf{m} = \mathbf{y}$[30]. The SHE of FM is also irrelevant because it is magnetization-independent by definition[28]. Thus, we consider SHE, AHE, and ISOC. Each process has the Onsager reciprocal process, which converts perpendicular spin current to in-plane charge current. Overall, therefore, nine processes (three charge-to-spin conversion processes by three inverse processes) are allowed (see Table 1).

We next explain the sign of $\Delta r_{MR}$ originating from each of nine processes: (1, 2) each of SHE[8–12] and AHE[26] causes a positive $\Delta r_{MR}$ when combined with its Onsager reciprocity. (3, 4) The sign of their cross-contributions depends on the sign of the product of the spin Hall angle of NM and the anomalous Hall angle of FM. Since solid symbols in Fig. 1h show the negativity of the anomalous Hall angle of NiFe ($\theta_{AH} = -0.0033$), it results in a positive $\Delta r_{MR}$ when multiplied by the negative spin Hall angle of Ta[5]. (5–8) There are four cross-contributions between the bulk effects (SHE and AHE) and ISOC. As we prove by a concise symmetry argument (Supplementary Note 7) as well as explicit calculations (Supplementary Note 8), they are canceled by the Onsager conjugates; for instance, a combined action of the SHE and inverse ISOC is exactly canceled out by that of the ISOC and inverse SHE. Therefore, the contributions (1–8) cannot contribute to the negative $\Delta r_{MR}$.

Among the nine processes, therefore, only one process remains; (9) a combined action of the ISOC and inverse ISOC. The analysis described in Supplementary Note 7 suggests that the negative $\Delta r_{MR}$ from this process is allowed by symmetry. For explicitly showing that this process contributes to the $\Delta r_{MR}$ negatively, we extend the previous spin drift-diffusion model[9,31], widely used to investigate charge and spin transport in NM/FM

bilayers, by including the interfacial charge-to-spin[21,22] and spin-to-charge conversion processes. We here show a simplified expression (see Supplementary Note 8 for the full expression), which neglects the longitudinal spin-current transmission between NM and FM layers (which is equivalent to the model A of ref. [11]). In this limit, the SHE and ISOC contributions to the $\Delta r_{MR}$ are given as

$$\Delta r_N^{SH} = \theta_{SH}^2 \frac{\rho_F l_{sf}^N}{\rho_F t_N + \rho_N t_F} \frac{\tilde{G} \tanh^2\left(\frac{t_N}{2 l_{sf}^N}\right)}{1 + \tilde{G} \coth\left(\frac{t_N}{l_{sf}^N}\right)}, \qquad (2)$$

$$\Delta r_N^{ISOC} = -\frac{\rho_F}{\rho_F t_N + \rho_N t_F} \left\{ \left[\sigma_{ISOC}^y(\mathbf{m} = \hat{\mathbf{y}})\right]^2 \left[\rho_N l_{sf}^N \coth\left(\frac{t_N}{l_{sf}^N}\right) + \rho_F l_{sf}^F \coth\left(\frac{t_F}{l_{sf}^F}\right)\right] \right.$$
$$\left. - \rho_N l_{sf}^N \frac{\left[\sigma_{ISOC}^y(\mathbf{m}=\hat{z})\right]^2 - \left[\sigma_{ISOC}^x(\mathbf{m}=\hat{z})\right]^2}{\tilde{G} + \tanh\left(\frac{t_N}{l_{sf}^N}\right)} \right\}. \qquad (3)$$

where $\theta_{SH}$ is the spin Hall angle of NM layer, $t_N$ is the thickness of NM, $l_{sf}^N$ ($l_{sf}^F$) is the spin-diffusion length of NM (FM) layer, $\rho_N$ ($\rho_F$) is the magnetization-independent resistivity of NM (FM) layer, $\tilde{G}(= 2\rho_N l_{sf}^N \text{Re}\left[G_{\uparrow\downarrow}\right])$ is the dimensionless mixing conductance, and $\sigma_{ISOC}^y(\mathbf{m})$ and $\sigma_{ISOC}^x(\mathbf{m})$ are the conductivities corresponding to the interfacial spin filtering and precession processes, respectively[21–23], both of which depend on $\mathbf{m}$. The first term in Eq. (3) is always negative and the second term is conditionally negative when $\sigma_{ISOC}^x(\mathbf{m} = \mathbf{z}) \geq \sigma_{ISOC}^y(\mathbf{m} = \mathbf{z})$. Because $\tilde{G}(= 2G_{\uparrow\downarrow} l_{sf}^N \rho_N)$ is proportional to the resistivity of NM layer, the first term of Eq. (3) would be dominant over the second term for highly resistive NM layers (such as Ta). In Fig. 3, we show the

calculation result of the MR ratio ($\Delta r_{MR}$) from various spin-charge conversion processes. Among four processes, the only one process by ISOC (9) contributes to the negative $\Delta r_{MR}$. The negative $\Delta r_{MR}$ from ISOC-related process can qualitatively account for our experimental results in Ta/NiFe sample. In the calculation, we use the model parameters of Ta/NiFe in Supplementary Note 8. Because the anomalous Hall angle of our NiFe sample is small ($\theta_{AH} = -0.0033$), the AHE-related $\Delta r_{MR}$ (2–4) is small.

**Negative SMR from the interfacial conversion process.** We qualitatively explain why the interfacial interconversion could result in the negative SMR (see Fig. 4 for a schematic explanation). For simplicity, we assume that $\sigma_{ISOC}$ is independent of **m** and is nonzero only for the spin-filtering component (i.e.,

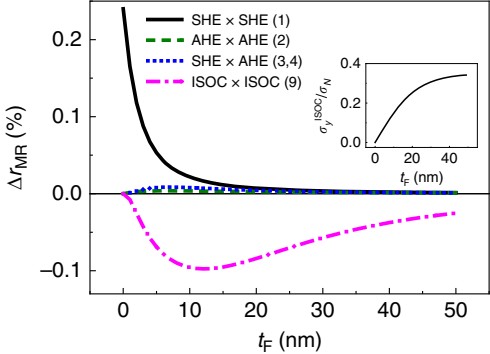

**Fig. 3 Calculated MR ratio ($\Delta r_{MR}$) from various spin-to-charge conversion processes.** SHE bulk spin Hall effect in NM, AHE anomalous Hall effect in FM, ISOC interfacial spin–orbit coupling effect at NM/FM interface. The numbers in parentheses correspond to those explained in the main text. The inset shows the interfacial spin–orbit coupling effect ($\sigma_{ISOC}^y$), which is used for the fitting of $\Delta r_{MR}$ from ISOC-related process (magenta dash dot line).

$\sigma_x^{ISOC} = 0$). Then, Eq. (3) becomes

$$\Delta r_N^{ISOC} = -\frac{\rho_F \rho_N l_{sf}^N}{\rho_F t_N + \rho_N t_F} \left(\sigma_{ISOC}^y\right)^2$$
$$\left[\coth\left(\frac{t_N}{l_{sf}^N}\right) + \frac{\rho_F l_{sf}^F}{\rho_{NN} l_{sf}^N}\coth\left(\frac{t_F}{l_{sf}^F}\right) - \frac{1}{\tilde{G} + \tanh\left(\frac{t_N}{l_{sf}^N}\right)}\right]. \quad (4)$$

We note that Eq. (4) is always negative because $\coth(x) > 1/[\tilde{G} + \tanh(x)]$, whenever $\tilde{G}$ is positive. Let us assume that the amount of spin current generated by either bulk or interface spin–orbit interaction is the same as $J_S$. Regardless of the spin-current source, the amount of spin-current absorption by FM depends on the direction of **m**, satisfying $J_S^{abs}(\mathbf{m} = \mathbf{z}) > J_S^{abs}(\mathbf{m} = \mathbf{y})$, because the spin angular momentum is transferred to the FM magnetization for $\mathbf{m} = \mathbf{z}$, but not for $\mathbf{m} = \mathbf{y}$. For the bulk SHE of NM, the spin current is reflected at the NM/FM interface, because it is generated away from the interface. The reflected spin current $J_S^{ref}$ follows $J_S^{ref}(\mathbf{m} = \mathbf{z}) < J_S^{ref}(\mathbf{m} = \mathbf{y})$ because $J_S = J_S^{abs} + J_S^{ref}$, as we neglect the longitudinal transmission of spin current through the interface. As a result, net spin current $J_S^{net}(\mathbf{m})$ $[=J_S - J_S^{ref}(\mathbf{m})]$ in bulk NM follows $J_S^{net}(\mathbf{m} = \mathbf{z}) > J_S^{net}(\mathbf{m} = \mathbf{y})$. Through the inverse SHE of bulk NM, this net spin current is converted into in-plane charge current, which increases the resistance of bilayer and results in $R(\mathbf{m} = \mathbf{z}) > R(\mathbf{m} = \mathbf{y})$, i.e., a positive SMR.

In contrast to the bulk contribution, the interface-generated spin current is not reflected at the interface because the interface itself is a spin-current source. For the ISOC, thus, net spin current $J_S^{net}$ at the NM/FM interface is given by $J_S^{net}(\mathbf{m}) = J_S - J_S^{abs}(\mathbf{m})$, which follows $J_S^{net}(\mathbf{m} = \mathbf{z}) < J_S^{net}(\mathbf{m} = \mathbf{y})$ because $J_S^{abs}(\mathbf{m} = \mathbf{z}) > J_S^{abs}(\mathbf{m} = \mathbf{y})$, resulting in a negative $\Delta R_{MR}$. As a remark, neglecting the longitudinal transmission to obtain Eqs. (2) and (3) does not alter our main conclusion, because the magnetoelectric circuit theory guarantees its upper bound by $2\text{Re}[G^{\uparrow\downarrow}] \geq G^{\uparrow} + G^{\downarrow}$ (Supplementary Note 8).

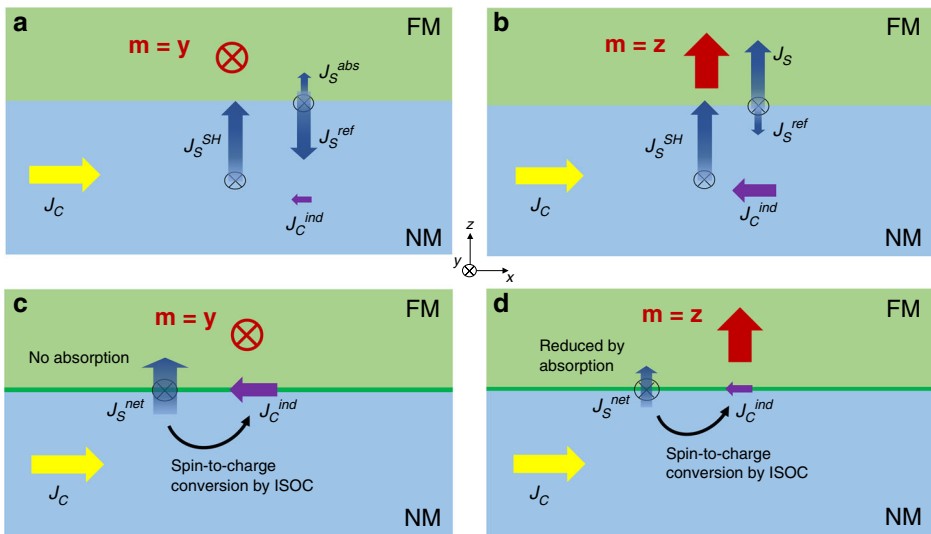

**Fig. 4 Schematic illustrations of charge-to-spin interconversion in NM/FM bilayers. a, b** Spin- and charge-current generations from bulk SHE and inverse SHE for **m||y** (**a**), and **m||z** (**b**). **c, d** Spin- and charge-current generations from ISOC and inverse ISOC effects for **m||y** (**c**), and **m||z** (**d**). $J_C$ and $J_C^{ind}$ are applied charge current and induced charge current by spin-to-charge conversion process, respectively. $J_S^{SH}$ is a spin current generated by bulk SHE of NM, $J_S^{abs}$ is a spin current absorbed by FM, $J_S^{ref}$ is a spin current reflected at the NM/FM inteface, and $J_S^{net}$ represents a net spin current.

To fit the experimental data with the model calculation, we use the extended spin drift-diffusion model, including the interfacial interconversion processes. In order to eliminate the negative $\Delta r_{MR}$ from GSE, we use the parallel circuit model approximation[15]. We note that our model calculation is only for qualitative description of $\Delta r_{MR}$ because the parallel circuit model is not reliable enough for quantitative calculation. However, the argument on the existence of additional source of the negative $\Delta r_{MR}$ is valid in our experiment as discussed in Supplementary Note 4. In Supplementary Note 8, we show the calculation details and the fitting result of $\Delta r_{MR}$ in the Ta/NiFe sample (see Supplementary Fig. 7). We obtain a good fitting with our model calculation, but quantitative understanding of the magnetoresistance demands further studies. Based on these results combined with the extended spin drift-diffusion model including the ISOC, we suggest approaches to observe the negative SMR in NM/FM bilayers at the end of Supplementary Note 8.

## Discussion

We have demonstrated negative SMR of Ta/NiFe bilayers, which is described by a combined action of the ISOC-induced charge-current-to-spin-current conversion, observed recently[23], and the ISOC-induced spin-current-to-charge-current conversion, first proven here. Although the Onsager reciprocity guarantees the coexistence of these two phenomena, the latter has not been demonstrated yet. In bulk, the inverse SHE has served as an important tool for research in spin transport in spin–orbit-coupled materials. Similarly, our demonstration of the ISOC-induced spin-current-to-charge-current conversion will invigorate research in spin phenomena near spin-orbit coupled interfaces, as the inverse SHE did. Moreover, the different signs of SMR depending on its origin will provide an unambiguous way to identify the different origins of spin–orbit coupling phenomena because interface and bulk contributions have the same symmetries and are uneasy to disentangle[19,20,31,32].

## Methods

**Sample preparation**. The samples of NiFe ($Ni_{81}Fe_{19}$, 1 ~50 nm) single layer and NM (Pt or Ta, 3 nm)/NiFe (1 ~ 50 nm) bilayer structures were deposited on thermally oxidized Si substrates by ultrahigh-vacuum magnetron sputtering with a base pressure of less than $4.0 \times 10^{-6}$ Pa ($=3.0 \times 10^{-8}$ Torr) at room temperature. On top of those structures, MgO (1.6 nm)/Ta (2 nm) layers were deposited to prevent natural oxidation. All metallic layers were grown by DC sputtering with a working pressure of 0.4 Pa ($=3$ mTorr), while the MgO layer was deposited by RF sputtering at 1.33 Pa ($=10$ mTorr). The Hall-bar-structured devices with a cross-structure of 5 μm × 35 μm were fabricated using photolithography and Ar ion-milling technique.

**Spin Hall magnetoresistance measurement**. The SMR was characterized by measuring the longitudinal resistance using a DC current of 100 μA in the $x$ direction while rotating an external magnetic field of 9 T in the $yz$ plane. Since 9 T is much larger than the anisotropy field of NiFe (~1 T), the magnetization is aligned to the direction of the applied magnetic field. All measurements were carried out at room temperature. More than three devices were measured for each type of sample; data are qualitatively reproducible. The statistical variation of the SMR is included as error bars in Fig. 1.

## Data availability

The data that support the findings of this study are available from the corresponding author upon reasonable request.

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

## Acknowledgements

We acknowledge discussion with V.P. Amin, P.H. Haney, and M.D. Stiles. K.-J.L. and B.G.P. acknowledge support from Samsung Research Funding Center of Samsung Electronics under Project Number SRFCMA1702-02. K.-W.K acknowledges support

from the KIST Institutional Program (Project Nos. 2V05750 and 2E30600). G.G. acknowledges a support by the NRF under Grant (NRF-2019R1I1A1A01063594).

## Author contributions

B.-G.P. and K.-J.L. planned and designed the experiment. M.-G.K., J.-G.C., and B.-G.P. prepared samples and performed the measurements. G.G., K.-W.K., and K.-J.L. provided the theory. All authors discussed the results and wrote on the paper.

## Competing interests

The authors declare no competing interests.
