## [Peer Review File · Nature Communications]

Reviewers' comments:

Reviewer #1 (Remarks to the Author):

The manuscript entitled "Negative spin Hall magnetoresistance of normal metal/ferromagnet bilayers" by Kang et al. reports negative SMR of Ta/NiFe bilayers, and they ascribe it to a combined action of the ISOC-induced charge-current-to-spin-current conversion. The sign of SMR for Ta/NiFe sample changes at $t_F = 2$ nm, whereas the sign for Pt/NiFe sample changes at $t_F = 10$ nm. They exclude the GSE as the origin of negative SMR and qualitatively explain why the interfacial interconversion could result in negative SMR. Although the result is interesting, I am afraid I cannot support the publication of this manuscript in its current form in Nature Communications. The interface-generated spin currents may exist in the NM/FM heterostructures. However, the manuscript suffers from severe problems and it is not reassuring that the authors indeed prove the existence of the interface-generated spin currents only through the sign change of SMR in the Ta/NiFe films. The related issues are elaborated in the attached word file. I cannot show it here because it includes figure and equations.

The manuscript entitled “Negative spin Hall magnetoresistance of normal metal/ferromagnet bilayers” by Kang *et al.* reports negative SMR of Ta/NiFe bilayers, and they ascribe it to a combined action of the ISOC-induced charge-current-to-spin-current conversion. The sign of SMR for Ta/NiFe sample changes at $t_F = 2$ nm, whereas the sign for Pt/NiFe sample changes at $t_F = 10$ nm. They exclude the GSE as the origin of negative SMR and qualitatively explain why the interfacial interconversion could result in negative SMR. Although the result is interesting, I am afraid I cannot support the publication of this manuscript in its current form in Nature Communications. The interface-generated spin currents may exist in the NM/FM heterostructures. However, the manuscript suffers from severe problems and it is not reassuring that the authors indeed prove the existence of the interface-generated spin currents only through the sign change of SMR in the Ta/NiFe films. The related issues are elaborated in the following:

1. The analysis of excluding GSE mechanisms presented by the authors is flawed for two reasons. (i) The resistance change in NM layer has not been proved to be positive due to bulk SHE in the films prepared by the authors. More importantly, I wonder why the authors have not discussed the influence from the top MgO/Ta layers, in which the thin MgO is only 1.6-nm-thick and the TaO_x layer may also generate spin current and transmit through the thin MgO layer into NiFe layer as discussed in [Nature Comm 7,10644 (2016)]. (ii) The different temperature dependence of MR cannot serve as sufficient condition to exclude the GSE.
2. It seems that the authors have agreed with the theoretical results in [PRL 121, 136805 (2018)], in which the spin currents strongly depend on the spin-orbit field. Therefore, the authors should show, for example in the Supplementary Information, the discussion of the spin-orbit field in different heterostructures, because the directions of interfacial spin-orbit fields depend on the Rashba parameters of MgO, Pt, Ta and NiFe layer. In this case, the interface-generated spin current may offset the SHE-generated due to their opposite spin orientation.

3. For the benefit of the reader, the authors should estimate and compare the fraction of the electrical current flowing through the Ta and Pt portions in their films. As discussed in [PRL 121, 136805 (2018)], the interface-generated spin currents scale with electron lifetimes (which are monotonically related to the conductivity) so the same ratio is largely independent of conductivity. These spin currents are therefore more likely to be important in high conductivity samples. Therefore, given the conductivity of Ta layer should be much smaller, why the negative SMR is more pronounced in Ta/NiFe films?
4. In Supplementary Note 4, the authors have used the linear response matrix $L_{CS} = (j_p m_z, j_f, 0)$ for the ISOC. However, both the spin-orbit filtering and spin-orbit precession currents should depend on the magnetizations as shown in the following figures in [PRL 121, 136805 (2018)]. Therefore, the authors cannot express the charge current originating from various charge-to-spin conversions and their inverse effects using the equation

$$(L_{CS}^{SHE} + L_{CS}^{ISOC}) \cdot (L_{SC}^{SHE} + L_{SC}^{ISOC}) = j_f^2 - j_p^2 m_z^2 - \sigma_{SH}^2$$

, in which the SHE does not depend on the magnetizations. Moreover, the spin-orbit precession currents seem to be more complicated, and the authors should discuss it separately.

[Redacted]

5. In Supplementary Note 4, the authors have used the following boundary conditions in the NM/FM structure as $j_z = 0$, $\mathbf{j}_z^s(z = -t_N) = 0$, $\mathbf{j}_z^s(z = t_F) = 0$, and I also suspect

whether the spin current is zero when $z=t_F$, especially for the thinner NiFe films.

6. According to comments 4 and 5, the authors should re-discuss and give the detailed expression of $\mathbf{j}_{s,T}^{ISOC}(0^-)$ and $j_{s,L}^{ISOC}(0^-)$ in equations (14) and (15) of Supplementary Note 4.
7. In its current form and according to my estimate for the SMR of single NiFe layer, the data does not support an interface-generated spin current origin of the negative SMR as claimed by the authors. The authors can also investigate the SMR in, for example, the Cu/NiFe bilayers, in which the SHE of Cu should be very weak.
8. The theoretical discussion of ISOC in [PRL 121, 136805 (2018)] and experimental results in [Nat. Mater. 17, 509 (2018)] also predicts a perpendicular spin orientation in the NM/FM bilayers, so what is its influence on the SMR?
9. Besides the SMR, have the authors carefully considered the spin-orbit torques measurements to further prove their speculation? They can carry out the harmonic measurements to investigate the interface-generated spin currents in, for example Cu/NiFe bilayers.
10. According to comments 7 and 9, the authors can give the detailed discussion of effect spin Hall angle due to the presence of interface-generated spin currents, for example what will determine its sign and magnitude? Actually, it has not been precisely described in its current form.

To conclude, the result reported by Kang *et al.* is interesting in the sense that it may indicate the emergence of interface-generated spin currents in NM/FM heterostructures. However the experiments and analyses presented is not convincing enough. The manuscript needs to be substantially modified before it can be considered for publication in Nature Communications.

Reviewer #2 (Remarks to the Author):

In this manuscript, the authors reported, for the first time, a negative spin Hall magnetoresistance (SMR) in a ferromagnetic-metal/heavy-metal bilayer (NiFe/Ta). By careful symmetry analysis and explicit model (drift-diffusion) calculation, they attributed the effect to a concerted action of interfacial spin-to-charge and charge-to-spin conversions. Overall, I find the result is novel and the data are technically sound, but a few points need to be clarified/modified (as stated below) before I can fully support the publication of the paper in Nature Communications.

1) Heavy-metal/ferromagnetic-metal bilayers are common systems to measure the SMR [see e.g., Ref.[11], PRB 87, 220409(R) (2013), PRL 106, 217207(2011)], and yet negative SMR had not been discovered before. It would be rather desirable for the authors to point out the conditions under which the effect can be observed [e.g., relations between thicknesses, diffusion lengths, spin Hall angles etc.]

2) The absorption of transverse spin current takes place at the interface of the bilayer within a very short length scale (typically a few angstroms to a nm) – sometimes known as spin-dephasing length. If the negative SMR originates from this interfacial spin current, then one would expect its characteristic length scale to go with the spin-dephasing length rather than the much longer spin diffusion length, just as the case of the inverse Edelstein-Rashba effect wherein the converted charge current is quasi-two-dimensional. It would be helpful to have this point clarified.

3) As the negative SMR was nailed down to the contribution from the interfacial Rashba spin-orbit coupling only, one would expect it to disappear (or at least significantly reduced) when a thin copper layer is inserted at the interface of the bilayer. Such control experiment will allow the authors to make their argument more convincing.

Reviewer #3 (Remarks to the Author):

The authors experimentally observed a negative spin Hall magnetoresistance (SMR), which is not compatible with the conventional theory for SMR. In the conventional theory, the SMR is always positive since it has a quadratic dependence on the spin Hall angle of the normal metal. Using a drift-diffusion model analysis, the authors attribute their experimental observation to the interfacial spin current that is converted to a charge current due to the interfacial spin orbit interaction. Moreover, the authors discussed several important conversion mechanisms of spin and charge currents to illustrate their contributions the longitudinal magnetoresistance. In these discussions, they concluded that only the interface spin orbit interaction could generate the negative SMR.

This is an interesting paper in the sense of the experimental observation and theoretical explanation. The physical picture proposed by the authors sounds reasonable. However, I still have some concerns about the correlation between the theory and the measurement, which are listed in detail below.

1 The authors seemed to focus only on the negative SMR while some other features are seen in their experimental data, for example, the positive magnetoresistance of Pt/NiFe at small t_F in Fig. 1g. The authors indeed mentioned competition of different mechanisms in the manuscript. But what could it be? Besides, the authors only fixed the thickness of 3 nm for Pt and Ta. If the interface spin orbit coupling is the key issue, what would we expect for the NM-thickness dependence? Is there a competition between the bulk and interface spin orbit interaction?

2 The authors ignored the interface Rashba induced MR because they believe that Rashba only contribute a positive MR. But I am not entirely sure about this statement. In fact, Rashba interaction may cause opposite splitting of two spins at the interface. Can the authors show that both possibilities give rise to the positive MR?

3 My last question is the statement about the difference in the interface spin accumulation and interface spin current (line 51-53). If there is a spin density at the interface, the imbalance of spin chemical potentials can drive a spin current through the interface, which is equivalent to an interface spin current perpendicular to the interface. Did I misunderstand anything? If so, I would suggest the authors provide a better explanation in the manuscript. This will be particularly important for non-experts of this field.

I would like to recommend the manuscript for publication in Nature Communications if the authors address the above questions/comments properly in a revised version.

Reviewer #1 (Remarks to the Author):

The manuscript entitled “Negative spin Hall magnetoresistance of normalmetal/ferromagnet bilayers” by Kang *et al.* reports negative SMR of Ta/NiFe bilayers, and they ascribe it to a combined action of the ISOC-induced charge-current-to-spin-current conversion. The sign of SMR for Ta/NiFe sample changes at $t_F = 2$ nm, whereas the sign for Pt/NiFe sample changes at $t_F = 10$ nm. They exclude the GSE as the origin of negative SMR and qualitatively explain why the interfacial interconversion could result in negative SMR. Although the result is interesting, I am afraid I cannot support the publication of this manuscript in its current form in Nature Communications. The interface-generated spin currents may exist in the NM/FM heterostructures. However, the manuscript suffers from severe problems and it is not reassuring that the authors indeed prove the existence of the interface-generated spin currents only through the sign change of SMR in the Ta/NiFe films. The related issues are elaborated in the following:

1. The analysis of excluding GSE mechanisms presented by the authors is flawed for two reasons. (i) The resistance change in NM layer has not been proved to be positive due to bulk SHE in the films prepared by the authors.

[Response] The SMR due to the bulk spin Hall effect (SHE) of non-magnet (NM) is known to be positive, which was well explained in H. Nakayama *et al.*, PRL 110, 206601 (2013) and Y.-T. Chen *et al.*, PRB **87**, 144411 (2013). Because of the bulk SHE of NM, a charge current along the x -direction generates a spin current travelling perpendicular to the film (\mathbf{z}) with a transverse spin polarization (\mathbf{y}). This spin current is absorbed and reflected at the FM/NM interface. Because the spin Hall current has the \mathbf{y} -spin polarization, the spin current absorption is maximized (minimized) for $\mathbf{m} = \mathbf{z}$ ($\mathbf{m} = \mathbf{y}$), while the spin current reflection is maximized (minimized) for $\mathbf{m} = \mathbf{y}$ ($\mathbf{m} = \mathbf{z}$). As the reflected spin current reduces net spin current in NM, the charge backflow induced by the inverse SHE of NM is smaller for $\mathbf{m} = \mathbf{y}$ than for $\mathbf{m} = \mathbf{z}$, resulting in a positive SMR (i.e., $\Delta\rho_2 > 0$). As the SMR for this mechanism is proportional to θ_{SH}^2 , where θ_{SH} is the spin Hall angle of NM, it is always positive regardless of the sign of θ_{SH} . The positive SMR due to bulk SHE of NM is described in the main text (on page 3, lines 37-43).

More importantly, I wonder why the authors have not discussed the influence from the top MgO/Ta layers, in which the thin MgO is only 1.6-nm-thick and the TaO_x layer may also generate spin current and transmit through the thin MgO layer into NiFe layer as discussed in [Nature Comm 7,10644 (2016)].

[Response] To verify the effect of the top MgO/Ta layer on the negative SMR observed in Ta/NiFe/MgO/Ta structures, we investigated two additional samples; one is a substrate/NiFe(5 nm)/Ta(3 nm)/MgO (1.6 nm)/Ta (2 nm) structure where the stacking order of Ta and NiFe is reversed, and the other is a Ta(3 nm)/NiFe(5 nm)/MgO(10 nm)/Ta (2 nm) structure where the MgO layer (10 nm) is thick enough to prevent spin current transmission from the top MgO/Ta interface. Figure R1 shows the SMR results of the two samples, compared to that of a reference sample of Ta(3 nm)/NiFe(5 nm)/MgO(1.6 nm)/Ta (2 nm) structure, demonstrating that SMR values are very similar within an error of 5% irrespective of the sample structures. This result suggests that the top MgO/Ta layer does not mainly contribute to the negative SMR in Ta/NiFe bilayers.

In revised Supplementary Note 2, we added the experimental results shown in Fig. R1.

Figure R1. SMR in various samples of Ta(3 nm)/NiFe(5 nm)/MgO(1.6 nm)/Ta, NiFe(5 nm)/Ta(3 nm)/MgO(1.6 nm)/Ta, and Ta(3 nm)/NiFe(5 nm)/MgO(10 nm)/Ta. SMR is defined as $\Delta R_{xx} = [R_{xx}(\beta = 0) - R_{xx}(\beta)]/R_{xx}(\beta = 0)$, where the magnetization angle β is defined in the schematic figure.

(ii) The different temperature dependence of MR cannot serve as sufficient condition to exclude the GSE.

[Response] We agree with the reviewer's comment so that we removed the temperature dependence of MR in the revised manuscript.

To further support our conclusion that the GSE alone cannot explain the negative SMR of the bilayers, we carried out an additional experiment for the NM thickness dependence of SMR in Ta/NiFe samples. We measured SMR in Ta(t_{Ta})/NiFe(5 nm) bilayers with Ta thickness t_{Ta} ranging from 1 nm to 12 nm. The measurement condition is the same as described in Method Section of the main text. As seen in Fig. R2a-b, the SMR is negative for all t_{Ta} 's, and becomes maximum when $t_{\text{Ta}}=1\sim 2$ nm. The GSE contribution from the NiFe(5 nm) layer (estimated from the parallel circuit model, $\Delta r_{MR}^{N/F} = \frac{R_0}{R_F} \Delta r_{MR}^F + \frac{R_0}{R_N} \Delta r_{MR}^N$ with fixed Δr_{MR}^F and $\Delta r_{MR}^N = 0$) is shown by a red line. Here R_F , R_N and R_0 are the magnetization-independent resistance of FM, NM, and NM/FM bilayer, respectively, and Δr_{MR}^F (Δr_{MR}^N) is the resistance change (in ratio) of FM (NM) layer. We find that the GSE alone cannot explain the negative SMR in Ta/NiFe bilayer, necessitating an additional source of the negative SMR in this sample. On the other hand, we could get a reasonable fit when considering the interfacial spin-orbit coupling (ISOC) effect, which is shown in a purple line in Fig. R2b.

In the revised main text, we added the experimental results shown in Fig. R2 as Fig. 2.

Figure R2. Ta-thickness dependence of SMR in Ta/NiFe. a, β -angle dependence of SMR [$\Delta R_{xx} = [R_{xx}(\beta = 0) - R_{xx}(\beta)]/R_{xx}(\beta = 0)$] in Ta(t_{Ta})/NiFe(5 nm). β is indicated in the left schematic. **b**, Ta-thickness (t_{Ta}) dependence of MR ratio (Δr_{MR}) of the Ta/NiFe sample. Solid lines are fitting results with including only GSE (red), and ISOC effect with GSE (purple).

(purple).

2. It seems that the authors have agreed with the theoretical results in [PRL 121, 136805 (2018)], in which the spin currents strongly depend on the spin-orbit field. Therefore, the authors should show, for example in the Supplementary Information, the discussion of the spin-orbit field in different heterostructures, because the directions of interfacial spin-orbit fields depend on the Rashba parameters of MgO, Pt, Ta and NiFe layer. In this case, the interface-generated spin current may offset the SHE-generated due to their opposite spin orientation.

[Response] As the referee pointed out, the direction of spin-orbit field and, equivalently, the sign of interfacial spin Hall conductivity (σ_{ISOC}), depend on the sign of the Rashba parameter at the interface. We note however that the sign of SMR is independent of the sign of the Rashba parameter because the SMR results from a combined action between the charge-to-spin conversion [proportional to the bulk (σ_{SH}) and interfacial (σ_{ISOC}) spin Hall conductivities] and its inverse effect [also proportional to the bulk (σ_{SH}) and interfacial (σ_{ISOC}) spin Hall conductivities]. Therefore, the SMR from the bulk SHE is proportional to σ_{SH}^2 and the interfacial spin-charge interconversion process is proportional to σ_{ISOC}^2 . As a result, the ISOC contribution to SMR does not depend on the sign of σ_{ISOC} and, equivalently, the direction of the spin-orbit field.

Concerning the cross contributions between the bulk SHE and ISOC, they are cancelled by the Onsager conjugates. In other words, a combined action of the SHE and inverse ISOC is exactly cancelled out by that of the ISOC and inverse SHE (see Supplementary Note 7 for a detailed symmetry argument and see Supplementary Note 8 for explicit calculations).

3. For the benefit of the reader, the authors should estimate and compare the fraction of the electrical current flowing through the Ta and Pt portions in their films. As discussed in [PRL 121, 136805 (2018)], the interface-generated spin currents scale with electron lifetimes (which are monotonically related to the conductivity) so the same ratio is largely independent of conductivity. These spin currents are therefore more likely to be important in high conductivity samples. Therefore, given the conductivity of Ta layer should be much smaller, why the negative SMR is more pronounced in Ta/NiFe films?

[Response] The measured resistivities are $\rho_{\text{Pt}} = 40 \mu\Omega\text{cm}$, $\rho_{\text{Ta}} = 300 \mu\Omega\text{cm}$, and $\rho_{\text{NiFe,5nm}} = 110 \mu\Omega\text{cm}$, respectively (Supplementary Note 3). The difference in resistivity between NM and FM induces non-uniform current distribution; 20% (66%) of the current flows through Ta (Pt) in Ta(Pt)/NiFe (5 nm) sample. We note that we have taken this non-uniform current distribution into account to calculate the SMR in the original manuscript.

We assume that the referee's comment (i.e., the interface-generated spin currents scale with electron lifetime) is based on Eqs. (4) and (5) of PRL **121**, 136805 (2018). We note that the authors of the PRL have simplified these equations too much in order to qualitatively describe the interface-generated spin currents. The simplification is evidenced by the fact that the electron lifetimes in Eqs. (4) and (5) are all spin-dependent and thus the contributions originating from the electron lifetime of the NM layer do not explicitly appear in the equations. An explicit description is given in Eqs. (S9) and (S10) of the Supplementary Material of Nat. Mater. **17**, 509 (2018), as shown below.

$$\text{Eq. (S9): } j_{\sigma}^I = C(\tau_{NM} - \tau_{FM}) \int d\bar{k}_x d\bar{k}_y \bar{k}_x T(\mathbf{k})_{\sigma C},$$

$$\text{Eq. (S10): } j_{\sigma}^{II} = C\tau_{FM} \int d\bar{k}_x d\bar{k}_y \bar{k}_x T(\mathbf{k})_{\sigma\sigma'} m_{\sigma'}.$$

Equation (S9) describes the spin-orbit filtering process and thus corresponds to Eq. (4) of the PRL, whereas Eq. (S10) describes the spin-orbit precession process and corresponds to Eq. (5) of the PRL. Equations (S9) and (S10) show that not only the electron lifetime of NM (τ_{NM}) but also that of FM (τ_{FM}) is important for the interface-generated spin currents. However, it is uneasy to estimate the magnitude of interface-generated spin current from the only electron lifetimes because it is also proportional to the Rashba strength at the interface, which is included in the scattering matrix $T(\mathbf{k})$ of Eqs. (S9) and (S10). It is known that the Rashba strength is strongly related to the work function difference across the interface [H. Tsai et al Sci. Rep. **8**, 5564 (2018)]. According to H. B. Michaelson [J. Appl. Phys. **48**, 4729 (1977)], the work function difference between Ta and Ni, $\phi_{\text{Ta-Ni}} (= 0.9 \text{ eV})$, is larger than work function difference between Pt and Ni, $\phi_{\text{Pt-Ni}} (= 0.5 \text{ eV})$. This larger $\phi_{\text{Ta-Ni}}$ than $\phi_{\text{Pt-Ni}}$ indicates that the Rashba parameter at the Ta/NiFe interface would be larger than that at the Pt/NiFe interface, which in turn generates a larger interfacial spin current in Ta-based samples.

4. In Supplementary Note 4, the authors have used the linear response matrix for the ISOC. However, both the spin-orbit filtering and spin-orbit precession currents should depend on the magnetizations as shown in the following figures in [PRL 121, 136805 (2018)]. Therefore, the authors cannot express the charge current originating from various charge-to-spin conversions and their inverse effects using the equation

$$(L_{CS}^{SHE} + L_{CS}^{ISOC}) \cdot (L_{SC}^{SHE} + L_{SC}^{ISOC}) = j_f^2 - j_p^2 m_z^2 - \sigma_{SH}^2,$$

in which the SHE does not depend on the magnetizations. Moreover, the spin-orbit precession currents seem to be more complicated, and the authors should discuss it separately.

[Response] As the referee pointed out, each of the coefficients appearing in our symmetry argument can depend on the magnetization direction. We note however that the symmetry argument is valid regardless of the dependence of coefficients on the magnetization direction. The only assumptions we made in the symmetry argument are the time-reversal properties of the coefficients, e.g., $\sigma_{SH}(\mathbf{m}) = \sigma_{SH}(-\mathbf{m})$. By noting that a spin current and an electric field are time-reversal even, the definition $\mathbf{j}_s = \sigma_{SH} E_x$ shows that σ_{SH} is also time-reversal even, i.e., $\sigma_{SH}(\mathbf{m}) = \sigma_{SH}(-\mathbf{m})$. In this way, one can specify the time-reversal properties of all the coefficients. Therefore, our symmetry argument is generally valid regardless of their \mathbf{m} dependencies.

5. In Supplementary Note 4, the authors have used the following boundary conditions in the NM/FM structure as $j_z = 0$, $\mathbf{j}_z^s(z = -t_N) = 0$, $\mathbf{j}_z^s(z = t_F) = 0$, and I also suspect whether the spin current is zero when $z = t_F$, especially for the thinner NiFe films.

[Response] As the referee pointed out, when t_F is small, a transverse spin current may survive at $z = t_F$, which modifies the boundary conditions in thin t_F cases. This modification will change the magnitude of SMR [Phys. Rev. B 96, 174412 (2017)]. However, we note that the sign of each SMR determined by each contribution is unchanged because the general symmetry argument in Supplementary Note 7 is valid regardless of the detailed boundary conditions. In addition, we observed a clear negative SMR signal for thicker t_F cases where the used boundary conditions are valid. Therefore, a possible modification of the boundary condition for thin t_F cases does not alter our main conclusion.

6. According to comments 4 and 5, the authors should re-discuss and give the detailed expression of and in equations (14) and (15) of Supplementary Note 4.

[Response] Reflecting this comment, we added the following sentences below Eq. (16) of the Supplementary Note 8.

“We note that these boundary conditions may need to be modified for thin NiFe films because a transverse spin current may survive at the outer surface of NiFe. This modification will change the magnitude of SMR. However, we note that the sign of each SMR determined by each contribution is unchanged because the general symmetry argument in Supplementary Note 7 is valid regardless of the detailed boundary conditions. In addition, we observed a clear negative SMR signal for thicker t_F cases where the used boundary conditions are valid. Therefore, a possible modification of the boundary condition for thin t_F cases does not alter our main conclusion.”

7. In its current form and according to my estimate for the SMR of single NiFe layer, the data does not support an interface-generated spin current origin of the negative SMR as claimed by the authors. The authors can also investigate the SMR in, for example, the Cu/NiFe bilayers, in which the SHE of Cu should be very weak.

[Response] We thank the reviewer for the suggestion to investigate SMR in Cu/NiFe bilayers, in which the bulk spin Hall effect is negligible. We fabricated Cu(t_{Cu})/NiFe(5 nm) bilayers with various Cu thicknesses t_{Cu} from 1 nm to 12 nm. Figure R3a shows the angle-dependent MR of Cu/NiFe bilayers, where the angle β is defined in the schematic. Figure R3b summarizes the SMR as a function of t_{Cu} . It is found that the SMR decreases with increasing t_{Cu} and is negative for all samples.

The overall tendency of decreased SMR with increasing t_{Cu} can be explained by the reduced GSE of NiFe due to the current shunting through the Cu layer. In order to check if the current shunting effect explains the observed t_{Cu} dependence of SMR quantitatively, we fit the experimental results with considering the only GSE, estimated from the parallel circuit model (a red curve in Fig. R3b). We find some deviations between the experimental results and fitting curve, implying that the Cu/NiFe interface may generate a spin current. This possibility of interface-generated spin current at the Cu/NiFe interface may be supported by the additional experiment of spin-orbit torque (SOT) measurements, demonstrating non-

negligible damping-like effective field (B_{DLT}) in Cu/NiFe bilayers. The experimental data for the SOT measurement will be presented in the response to Question #9 below.

In the revised Supplementary Note 5, we added the experimental results shown in Fig R3.

Figure R3. Cu-thickness dependence of SMR in Cu/NiFe. a, β -angle dependence of MR [$\Delta R_{xx} = [R_{xx}(\beta = 0) - R_{xx}(\beta)]/R_{xx}(\beta = 0)$] in Cu(t_{Cu})/NiFe(5). β is indicated in the left schematic. **b,** Cu thickness (t_{Cu}) dependence of SMR ratio (Δr_{MR}). Solid red curve describes fitting result with only considering GSE of NiFe.

8. The theoretical discussion of ISOC in [PRL 121, 136805 (2018)] and experimental results in [Nat. Mater. 17, 509 (2018)] also predicts a perpendicular spin orientation in the NM/FM bilayers, so what is its influence on the SMR?

[Response] In PRL 121, 136805 (2018), the interface-generated spin current is given as

$$\mathbf{j}_{interface} = j_f \hat{\mathbf{S}} + j_p \hat{\mathbf{m}} \times \hat{\mathbf{S}} + j_m \hat{\mathbf{S}} \times (\hat{\mathbf{m}} \times \hat{\mathbf{S}}).$$

For an external electric field \mathbf{E} applied along the x -direction, the spin polarization direction $\hat{\mathbf{S}}$ is in the y -direction (i.e., $\hat{\mathbf{S}} = \hat{\mathbf{z}} \times \hat{\mathbf{E}} = \hat{\mathbf{y}}$). Because j_m vanishes when $\hat{\mathbf{m}}$ lies in the plane or points out of the plane, we neglect the last term. Then, the z -polarized spin current is generated by the spin-orbit precession term (i.e., $j_p \hat{\mathbf{m}} \times \hat{\mathbf{S}}$) for the x -component of magnetization. As a result, the z -polarized spin current originating from the spin-orbit precession does not affect the SMR because the SMR measures the difference in MR between $\mathbf{m} = \hat{\mathbf{z}}$ and $\mathbf{m} = \hat{\mathbf{y}}$ (i.e., no contribution from the x -component of magnetization).

9. Besides the SMR, have the authors carefully considered the spin-orbit torques measurements to further prove their speculation? They can carry out the harmonic measurements to investigate the interface-generated spin currents in, for example Cu/NiFe bilayers.

[Response] Thanks to this comment, we measured current-induced SOT in Ta(3 nm)/NiFe(5 nm), NiFe (5 nm), and Cu(3 nm)/NiFe(5 nm) samples using in-plane harmonic Hall measurements [C. O. Avci *et al.*, Phys. Rev. B 90, 224427 (2014)]. As schematically depicted in top right of Fig. R4, the 1st and 2nd harmonic Hall resistances ($R_{xy}^{1\omega}, R_{xy}^{2\omega}$) are measured by rotating the sample (azimuthal angle φ) under a fixed in-plane magnetic field strength B_{ext} and an AC current I_{AC} . The $R_{xy}^{2\omega}$ is expressed as

$$R_{xy}^{2\omega}(\varphi) = \left(R_{\text{AHE}} \frac{B_{\text{DLT}}}{B_{\text{eff}}} + R_{\text{VT}}^{2\omega} \right) \cos\varphi + 2R_{\text{PHE}} \frac{B_{\text{FLT}} + B_{\text{Oe}}}{B_{\text{ext}}} (2\cos^3\varphi - \cos\varphi).$$

Here, R_{AHE} and R_{PHE} are the anomalous Hall and planar Hall resistances, respectively; B_{DLT} , B_{FLT} , and B_{Oe} are the damping-like effective field, the field-like effective field, the Oersted field, respectively; B_{eff} is the effective magnetic field including the demagnetization field B_{dem} and the anisotropy field B_{ani} of FM ($B_{\text{eff}} = B_{\text{ext}} + B_{\text{dem}} - B_{\text{ani}}$); $R_{\text{VT}}^{2\omega}$ is the thermoelectric contribution to $R_{xy}^{2\omega}$. Figures R4a-l show the measurement results of $R_{xy}^{1\omega}$ and $R_{xy}^{2\omega}$ versus φ curves for Ta/NiFe, NiFe, and Cu/NiFe samples. The $R_{xy}^{1\omega}$ exhibits a typical 2φ dependence due to the planar Hall effect (Fig. 4a-c), while the $R_{xy}^{2\omega}$ has both $\cos\varphi$ and $(2\cos^3\varphi - \cos\varphi)$ components (Fig. 4d-f), which are separately plotted in Fig. 4g-i and Fig. 4j-l, respectively. We extract the B_{DLT} from the slope of $\cos\varphi$ component of $R_{xy}^{2\omega}$ versus $1/B_{\text{eff}}$ curves (Fig. R4m). It is observed that the Ta/NiFe bilayer exhibits a sizable slope, resulting in B_{DLT} of ~ -0.45 mT/ $(1.0 \times 10^7 \text{ A/m}^2)$. On the other hand, the NiFe single layer shows a negligible slope and thereby vanishingly small B_{DLT} . Interestingly, the Cu/NiFe sample also exhibits a noticeable slope, but its sign is opposite to that of the Ta/NiFe sample, resulting in B_{DLT} of $\sim +0.32$ mT/ $(1.0 \times 10^7 \text{ A/m}^2)$. This demonstrates that a spin current is generated even in Cu/NiFe bilayer, which we attribute to the ISOC effect.

In the revised Supplementary Note 6, we added the experimental results shown in Fig R4.

Figure R4. Harmonic Hall resistance measurements in Ta/NiFe, NiFe and Cu/NiFe. Top right schematic indicates the detail measurement configuration. **a-c**, $R_{xy}^{1\omega}$ versus ϕ curves for Ta/NiFe (a), NiFe (b), and Cu/NiFe (c). The solid lines are $\sin 2\phi$ fits to the experimental data. **d-f**, $R_{xy}^{2\omega}$ versus ϕ curves for Ta/NiFe (d), NiFe (e), and Cu/NiFe (f). The solid lines are $\cos\phi + (2\cos^3\phi - \cos\phi)$ fits to the experimental data. **g-i**, The $\cos\phi$ component of $R_{xy}^{2\omega}$ versus ϕ curves for Ta/NiFe (g), NiFe (h), and Cu/NiFe (i). **j-l**, The $(2\cos^3\phi - \cos\phi)$ components of $R_{xy}^{2\omega}$ versus ϕ curves for Ta/NiFe (j), NiFe (k), and Cu/NiFe (l). **m**, The $\cos\phi$ component of $R_{xy}^{2\omega}$ versus $1/B_{\text{eff}}$ curves.

10. According to comments 7 and 9, the authors can give the detailed discussion of effect spin Hall angle due to the presence of interface-generated spin currents, for example what will determine its sign and magnitude? Actually, it has not been precisely described in its current form.

[Response] From the 2nd harmonic measurement summarized in Fig. R4, we obtain $\theta_{\text{DL}}^{\text{Ta/NiFe}} = -0.073$ in Ta(3 nm)/NiFe(5 nm) and $\theta_{\text{DL}}^{\text{Cu/NiFe}} = +0.037$ in Cu(3 nm)/NiFe(5 nm). The non-negligible $\theta_{\text{DL}}^{\text{Cu/NiFe}}$ is attributed to the interface-generated spin currents at the Cu/NiFe interface, because the bulk spin Hall effect of Cu is known to be negligible. On the

other hand, $\theta_{DL}^{\text{Ta/NiFe}}$ includes both contributions from the bulk spin Hall effect of Ta and the interface-generated spin current effect at the Ta/NiFe interface, which are uneasy to determine separately. One can estimate the effective spin Hall angle for the ISOC effect at the Ta/NiFe interface by fitting the negative SMR data with assuming the bulk spin Hall angle of Ta, as we have done in the main text. However, because these two effective spin Hall angles are entangled, we do not argue that the estimated value is quantitatively accurate.

To conclude, the result reported by Kang *et al.* is interesting in the sense that it may indicate the emergence of interface-generated spin currents in NM/FM heterostructures. However the experiments and analyses presented is not convincing enough. The manuscript needs to be substantially modified before it can be considered for publication in Nature Communications.

[Response] We hope our responses to the comments successfully resolve the reviewer's concern and the revised manuscript is now acceptable for publication.

Reviewer #2 (Remarks to the Author):

In this manuscript, the authors reported, for the first time, a negative spin Hall magnetoresistance (SMR) in a ferromagnetic-metal/heavy-metal bilayer (NiFe/Ta). By careful symmetry analysis and explicit model (drift-diffusion) calculation, they attributed the effect to a concerted action of interfacial spin-to-charge and charge-to-spin conversions. Overall, I find the result is novel and the data are technically sound, but a few points need to be clarified/modified (as stated below) before I can fully support the publication of the paper in Nature Communications.

[Response] We appreciate the reviewer's comment that "*the result is novel and the data are technically sound.*" We hope our responses to the comments successfully resolve the reviewer's concern and the revised manuscript is now acceptable for publication.

1) Heavy-metal/ferromagnetic-metal bilayers are common systems to measure the SMR [see e.g., Ref. [11], PRB 87, 220409(R) (2013), PRL 106, 217207(2011)], and yet negative SMR had not been discovered before. It would be rather desirable for the authors to point out the conditions under which the effect can be observed [e.g., relations between thicknesses, diffusion lengths, spin Hall angles etc.]

[Response] We suggest following approaches to observe the negative SMR in NM/FM bilayers:

One way is to maximize the ISOC contribution. Based on the spin drift-diffusion model, we obtained the ISOC contribution to SMR as

$$\Delta r_N^{ISOC} = -\frac{\rho_F}{\rho_F t_N + \rho_N t_F} \left\{ [\sigma_{ISOC}^y(\mathbf{m} = \hat{y})]^2 \left[\rho_N l_{sf}^N \coth\left(\frac{t_N}{l_{sf}^N}\right) + \rho_F l_{sf}^F \coth\left(\frac{t_F}{l_{sf}^F}\right) \right] - \rho_N l_{sf}^N \frac{[\sigma_{ISOC}^y(\mathbf{m} = \hat{z})]^2 - [\sigma_{ISOC}^x(\mathbf{m} = \hat{z})]^2}{\tilde{G} + \tanh\left(\frac{t_N}{l_{sf}^N}\right)} \right\}.$$

As clearly seen in the equation, the negative SMR increases in magnitude with increasing σ_{ISOC}^y (i.e., spin-orbit filtering contribution) and/or σ_{ISOC}^x (i.e., spin-orbit precession

contribution). As σ_{ISOC} increases with the Rashba constant, which is known to be strongly related to the work function difference across the interface [H. Tsai et al Sci. Rep. **8**, 5564 (2018)], a possible way to increase the negative SMR would be to use NM/FM bilayers with a large difference in the work function between NM and FM.

Another way is to maximize the ISOC contribution is to increase the spin diffusion length of NM (l_{sf}^N) and FM (l_{sf}^F), because the above equation shows that the negative SMR in bilayers with fixed t_N and t_F monotonically increases in magnitude with l_{sf}^N and l_{sf}^F .

An alternative way to increase the negative SMR is to use a NM layer with a high resistivity. It is because a less current flows through the NM layer as the resistivity of NM increases. It results in the reduction of the positive SMR originating from the bulk spin Hall effect of NM.

In the revised manuscript, we included these approaches to observe the negative SMR in NM/FM bilayers at the end of Supplementary Note 8.

2) The absorption of transverse spin current takes place at the interface of the bilayer within a very short length scale (typically a few angstroms to a nm) – sometimes known as spin-dephasing length. If the negative SMR originates from this interfacial spin current, then one would expect its characteristic length scale to go with the spin-dephasing length rather than the much longer spin diffusion length, just as the case of the inverse Edelstein-Rashba effect wherein the converted charge current is quasi-two-dimensional. It would be helpful to have this point clarified.

[Response] As the reviewer pointed out, the absorption of transverse spin current by FM takes place in the spin-dephasing length. A similar interface-related length scale may also exist in the NM layer, which is however not well established yet. A first-principles calculation [PRL **116**, 196602 (2016)] showed a giant interface spin Hall and inverse spin Hall effects in a NM/FM bilayer. Figure 3 of the PRL work (Fig. R5 below) showed that the interface spin current decays in NM within a very short length of about 1 nm, which is however not the spin diffusion length of NM (~ 5.6 nm, which was calculated by the same method in the PRL). We speculate this very short length scale is the characteristic length of interface-generated spin current in the NM layer, but a further study is required to investigate

its detail. As this is just a speculation for the moment, we decided not to add the above discussion in the manuscript.

[Redacted]

Figure R5. Figure 3 of PRL 116, 196602 (2016). Spin current in NM layer where the NM/FM interface is located at $z = 0$. One finds that a very large increase in the spin current near the interface of which length scale is about 1 nm.

3) As the negative SMR was nailed down to the contribution from the interfacial Rashba spin-orbit coupling only, one would expect it to disappear (or at least significantly reduced) when a thin copper layer is inserted at the interface of the bilayer. Such control experiment will allow the authors to make their argument more convincing.

[Response] Thanks to this comment, we performed an additional experiment of investigating the effect of Cu insertion layer on the SMR in a Ta(3 nm)/NiFe(5 nm) bilayer. Figure R6 shows the SMR of a Ta(3 nm)/Cu(1 nm)/NiFe(5 nm) sample. The Cu insertion significantly reduces the negative SMR, as expected by the reviewer. For comparison, we also measured SMR of Cu(3 nm)/NiFe (5 nm) bilayer (blue symbols in Fig. R6), which shows a similar reduction of the negative SMR. We attribute the non-zero SMR in Ta(3 nm)/Cu(1 nm)/NiFe(5 nm) and Cu(3 nm)/NiFe (5 nm) to the interfacial spin current generated at Cu/NiFe interface.

Figure R6. The influence of Cu insertion at the interface of Ta/NiFe bilayer. SMR in Ta(3 nm)/NiFe(5 nm) [black symbols], Ta(3 nm)/Cu(1 nm)/NiFe(5 nm) [red symbols], and Cu(3 nm)/NiFe(5 nm) [blue symbols]. SMR is defined as $\Delta R_{xx} = [R_{xx}(\beta = 0) - R_{xx}(\beta)]/R_{xx}(\beta = 0)$, where β is indicated in the right schematic.

To further support the ISOC origin of the negative SMR, we performed two additional experiments of SMR and spin-orbit torque measurements in Cu/NiFe bilayers. Figure R7a shows the angle-dependent MR of Cu(t_{Cu})/NiFe bilayers with various Cu thicknesses t_{Cu} from 1 nm to 12 nm, where the angle β is defined in the schematic. Figure R7b summarizes the SMR as a function of t_{Cu} . It is found that the SMR decreases with increasing t_{Cu} and is negative for all samples. The overall tendency of decreased SMR with increasing t_{Cu} can be explained by the reduced GSE of NiFe due to the current shunting through the Cu layer. In order to check if the current shunting effect explains the observed t_{Cu} dependence of SMR quantitatively, we fit the experimental results with considering the only GSE, estimated from the parallel circuit model (a red curve in Fig. R7b). We find some deviations between the experimental results and fitting curve, implying that the Cu/NiFe interface may generate a spin current. This possibility of interface-generated spin current at the Cu/NiFe interface may be supported by the spin-orbit torque (SOT) measurements, which is shown below.

In the revised Supplementary Note 5, we added the experimental results shown in Fig

R7.

Figure R7. Cu-thickness dependence of SMR in Cu/NiFe. a, β -angle dependence of MR [$\Delta R_{xx} = [R_{xx}(\beta = 0) - R_{xx}(\beta)]/R_{xx}(\beta = 0)$] in $\text{Cu}(t_{\text{Cu}})/\text{NiFe}(5)$. β is indicated in the left schematic. **b**, Cu thickness (t_{Cu}) dependence of SMR ratio (Δr_{MR}). Solid red curve describes fitting result with only considering GSE of NiFe.

We measured current-induced SOT in Ta(3 nm)/NiFe(5 nm), NiFe (5 nm), and Cu(3 nm)/NiFe(5 nm) samples using in-plane harmonic Hall measurements [C. O. Avci *et al.*, Phys. Rev. B 90, 224427 (2014)]. As schematically depicted in top right of Fig. R8, the 1st and 2nd harmonic Hall resistances ($R_{xy}^{1\omega}, R_{xy}^{2\omega}$) are measured by rotating the sample (azimuthal angle φ) under a fixed in-plane magnetic field strength B_{ext} and an AC current I_{AC} . The $R_{xy}^{2\omega}$ is expressed as

$$R_{xy}^{2\omega}(\varphi) = \left(R_{\text{AHE}} \frac{B_{\text{DLT}}}{B_{\text{eff}}} + R_{\nabla T}^{2\omega} \right) \cos\varphi + 2R_{\text{PHE}} \frac{B_{\text{FLT}} + B_{\text{Oe}}}{B_{\text{ext}}} (2\cos^3\varphi - \cos\varphi).$$

Here, R_{AHE} and R_{PHE} are the anomalous Hall and planar Hall resistances, respectively; B_{DLT} , B_{FLT} , and B_{Oe} are the damping-like effective field, the field-like effective field, the Oersted field, respectively; B_{eff} is the effective magnetic field including the demagnetization field B_{dem} and the anisotropy field B_{ani} of FM ($B_{\text{eff}} = B_{\text{ext}} + B_{\text{dem}} - B_{\text{ani}}$); $R_{\nabla T}^{2\omega}$ is the thermoelectric contribution to $R_{xy}^{2\omega}$. Figures R8a-l show the measurement results of $R_{xy}^{1\omega}$ and $R_{xy}^{2\omega}$ versus φ curves for Ta/NiFe, NiFe, and Cu/NiFe samples. The $R_{xy}^{1\omega}$ exhibits a typical 2φ dependence due to the planar Hall effect (Fig. 8a-c), while the $R_{xy}^{2\omega}$ has both $\cos\varphi$ and $(2\cos^3\varphi - \cos\varphi)$ components (Fig. 8d-f), which are separately plotted in Fig. 8g-i and Fig. 8j-l, respectively. We extract the B_{DLT} from the slope of $\cos\varphi$ component of

$R_{xy}^{2\omega}$ versus $1/B_{\text{eff}}$ curves (Fig. R8m). It is observed that the Ta/NiFe bilayer exhibits a sizable slope, resulting in B_{DLT} of $\sim -0.45 \text{ mT}/(1.0 \times 10^7 \text{ A/m}^2)$. On the other hand, the NiFe single layer shows a negligible slope and thereby vanishingly small B_{DLT} . Interestingly, the Cu/NiFe sample also exhibits a noticeable slope, but its sign is opposite to that of the Ta/NiFe sample, resulting in B_{DLT} of $\sim +0.32 \text{ mT}/(1.0 \times 10^7 \text{ A/m}^2)$. This demonstrates that a spin current is generated even in Cu/NiFe bilayer, which we attribute to the ISOC effect.

In the revised Supplementary Note 6, we added the experimental results shown in Fig. R8.

Figure R8. Harmonic Hall resistance measurements in Ta/NiFe, NiFe and Cu/NiFe. Top right schematic indicates the detail measurement configuration. a-c, $R_{xy}^{1\omega}$ versus φ curves for Ta/NiFe (a), NiFe (b), and Cu/NiFe (c). The solid lines are $\sin 2\varphi$ fits to the experimental data. d-f, $R_{xy}^{2\omega}$ versus φ curves for Ta/NiFe (d), NiFe (e), and Cu/NiFe (f). The solid lines are $\cos\varphi + (2\cos^3\varphi - \cos\varphi)$ fits to the experimental data. g-i, The $\cos\varphi$ component of $R_{xy}^{2\omega}$ versus φ curves for Ta/NiFe (g), NiFe (h), and Cu/NiFe (i). j-l, The $(2\cos^3\varphi - \cos\varphi)$ components of $R_{xy}^{2\omega}$ versus φ curves for Ta/NiFe (j), NiFe (k), and Cu/NiFe (l). m, The $\cos\varphi$ component of $R_{xy}^{2\omega}$ versus $1/B_{\text{eff}}$ curves.

Reviewer #3 (Remarks to the Author):

The authors experimentally observed a negative spin Hall magnetoresistance (SMR), which is not compatible with the conventional theory for SMR. In the conventional theory, the SMR is always positive since it has a quadratic dependence on the spin Hall angle of the normal metal. Using a drift-diffusion model analysis, the authors attribute their experimental observation to the interfacial spin current that is converted to a charge current due to the interfacial spin orbit interaction. Moreover, the authors discussed several important conversion mechanisms of spin and charge currents to illustrate their contributions the longitudinal magnetoresistance. In these discussions, they concluded that only the interface spin orbit interaction could generate the negative SMR.

This is an interesting paper in the sense of the experimental observation and theoretical explanation. The physical picture proposed by the authors sounds reasonable. However, I still have some concerns about the correlation between the theory and the measurement, which are listed in detail below.

[Response] We appreciate the reviewer's comment that *"This is an interesting paper in the sense of the experimental observation and theoretical explanation. The physical picture proposed by the authors sounds reasonable."* We hope our responses given below successfully resolve the reviewer's concern and the revised manuscript is now acceptable for publication.

1. The authors seemed to focus only on the negative SMR while some other features are seen in their experimental data, for example, the positive magnetoresistance of Pt/NiFe at small t_F in Fig. 1g. The authors indeed mentioned competition of different mechanisms in the manuscript. But what could it be?

[Response] The positive SMR of Pt/NiFe at small t_F means that the contribution of bulk spin Hall effect of Pt dominates the SMR in this bilayer. After excluding the GSE, which is a known mechanism to give a negative SMR, the SMR of Pt/NiFe becomes positive in entire tested t_F ranges. However, it does not mean that there is no ISOC effect at the Pt/NiFe interface. As the contributions from bulk spin Hall effect of NM and from ISOC effect are entangled, we are unable to quantitatively separate these two contributions. On the other hand, the negative SMR of Ta/NiFe even after excluding the GSE unambiguously evidences that

there is an additional source of SMR, which gives a negative sign. From our theoretical study with the symmetry argument, we conclude that the negative SMR arises from the interface-generated spin current.

Besides, the authors only fixed the thickness of 3 nm for Pt and Ta. If the interface spin orbit coupling is the key issue, what would we expect for the NM-thickness dependence? Is there a competition between the bulk and interface spin orbit interaction?

[Response] Thanks to this comment, we carried out an additional experiment for the NM thickness dependence of SMR in Ta/NiFe samples. We measured SMR in Ta(t_{Ta})/NiFe(5 nm) bilayers with Ta thickness t_{Ta} ranging from 1 nm to 12 nm. The measurement condition is the same as described in Method Section of the main text. As seen in Fig. R9a-b, the SMR is negative for all t_{Ta} 's, and it becomes maximum when $t_{\text{Ta}}=1\sim 2$ nm. The GSE contribution from the NiFe(5 nm) layer (estimated from the parallel circuit model, $\Delta r_{MR}^{N/F} = \frac{R_0}{R_F} \Delta r_{MR}^F + \frac{R_0}{R_N} \Delta r_{MR}^N$ with fixed Δr_{MR}^F and $\Delta r_{MR}^N = 0$) is shown in a red line. Here R_F , R_N and R_0 are the magnetization-independent resistance of FM, NM, and NM/FM bilayer, respectively, and Δr_{MR}^F (Δr_{MR}^N) is the resistance change (in ratio) of FM (NM) layer. We find that the GSE alone cannot explain the negative SMR in Ta/NiFe bilayer, necessitating an additional source of the negative SMR in this sample. On the other hand, we could get a reasonable fit when considering the interfacial spin-orbit coupling (ISOC) effect, which is shown in a purple line in Fig. R9b.

The negative SMR with excluding the GSE contribution implies that the contribution of ISOC at the Ta/NiFe interface dominates that of bulk spin Hall effect of Ta. However, we are unable to quantitatively separate these two contributions because they are entangled.

In the revised manuscript, we added the experimental results shown in Fig. R9 as Fig. 2.

Figure R9. Ta-thickness dependence of SMR in Ta/NiFe. **a**, β -angle dependence of SMR [$\Delta R_{xx} = [R_{xx}(\beta = 0) - R_{xx}(\beta)]/R_{xx}(\beta = 0)$] in Ta(t_{Ta})/NiFe(5 nm). β is indicated in the left schematic. **b**, Ta-thickness (t_{Ta}) dependence of MR ratio (Δr_{MR}) of Ta/NiFe sample. Solid lines are fitting results with including only GSE (red), and ISOC effect with GSE (purple).

2. The authors ignored the interface Rashba induced MR because they believe that Rashba only contribute a positive MR. But I am not entirely sure about this statement. In fact, Rashba interaction may cause opposite splitting of two spins at the interface. Can the authors show that both possibilities give rise to the positive MR?

[Response] We show below that the interface Rashba induced SMR is always positive. In this mechanism, the Rashba effect induces an additional velocity \mathbf{v}_{add}^{\pm} in the $\mathbf{M} \times \mathbf{z}$ direction:

$$\mathbf{v}_{add}^{\pm} \sim \eta \mathbf{M} \times \mathbf{z},$$

where η is the Rashba interaction [see V. L. Grigoryan, *et al.*, PRB **90**, 161412(R) (2014) and K. Narayanapillai *et al.*, PRB **96**, 064401 (2017)]. By inserting the additional velocity into

$$\sigma_{xx} \sim \int d^2k (v_x)^2 \delta(E_n - E_F),$$

one obtains the magnetization-dependent additional current along the applied electric field direction (\mathbf{x}):

$$J_x^{add} \sim \eta^2 M_y^2 \sigma_0 E_x.$$

Because the additional current positively maximizes for $\mathbf{M} = \mathbf{y}$, the intrinsic Rashba MR is always positive, same as the bulk spin Hall effect of NM. We note that the intrinsic Rashba MR is proportional to η^2 so that it is always positive regardless of the sign of η (i.e., the sign of Rashba splitting).

In the revised manuscript (on page 3, lines 44 – 49), we provided detailed explanation for positive SMR from the intrinsic Rashba MR.

3. My last question is the statement about the difference in the interface spin accumulation and interface spin current (line 51-53). If there is a spin density at the interface, the imbalance of spin chemical potentials can drive a spin current through the interface, which is equivalent to an interface spin current perpendicular to the interface. Did I misunderstand anything? If so, I would suggest the authors provide a better explanation in the manuscript. This will be particularly important for non-experts of this field.

[Response] As the referee commented, the interface spin accumulation from Rashba-Edelstein effect (REE) can also generate a spin current. However, it is qualitatively different from the interface-generated spin current. The former (REE) induces an imbalance of the spin chemical potential which generates a spin current by the spin diffusion. On the other hand, the latter directly generates a spin current from the interfacial spin-orbit scattering (filtering and precession), which gives different scattering amplitudes depending on the relative orientation between conduction electron spin and interfacial spin-orbit field. For example, in Y. Du *et al.* arXiv:1807.10867, the authors derived the SMR in the presence of the REE and bulk spin Hall effect of NM (thus, ignored the interface-generated spin current). The REE-induced SMR is given by

$$\Delta r_{MR} \sim \frac{\rho_N l_{sf}^N}{\rho_F t_N + \rho_N t_F} \left[\theta_{SH} \tanh\left(\frac{t_N}{2l_{sf}^N}\right) + \frac{\lambda_{IEE}}{2l_{sf}^N} \Phi \right]^2 \operatorname{Re} \left[\frac{g_s}{1 + \coth\left(\frac{t_N}{l_{sf}^N}\right) g_s} \right],$$

where the term with λ_{IEE} represents spin accumulation from the REE and g_s is the dimensionless mixing conductance (see Supplementary Note 4 of Y. Du *et al.* arXiv:1807.10867). As seen from the equation, the REE-induced SMR is always positive,

same as the bulk spin Hall effect of NM.

In the revised manuscript (on page 3, lines 51 – 57), we provided detailed explanation for the difference between the Rashba-Edelstein effect (REE) and the interface spin current.

I would like to recommend the manuscript for publication in Nature Communications if the authors address the above questions/comments properly in a revised version.

[Response] We hope our responses to the comments successfully resolve the reviewer's concern and the revised manuscript is now acceptable for publication.

REVIEWERS' COMMENTS:

Reviewer #1 (Remarks to the Author):

The authors have addressed all of my concerns. I think the current version is suitable to be accepted for the publication in Nature communications.

Reviewer #2 (Remarks to the Author):

No further comments.

Reviewer #3 (Remarks to the Author):

The authors have improved the manuscript according to the referees' reports. The experimental observation of the interface spin Hall effect via the negative spin-Hall magnetoresistance is indeed an interesting and important accomplishment in spintronics. The presentation of the manuscript is accessible for general readers. So I would like to recommend the manuscript for publication in Nature Communications.